# Antioxidant, Hypolipidemic and Hepatic Protective Activities of Polysaccharides from *Phascolosoma esculenta*

**DOI:** 10.3390/md18030158

**Published:** 2020-03-12

**Authors:** Yaqing Wu, Hongying Jiang, Jyuan-Siou Lin, Jia Liu, Chang-Jer Wu, Ruian Xu

**Affiliations:** 1Institute of Molecular Medicine, School of Biomedical Sciences & School of Medicine, Huaqiao University, Quanzhou 362021, China; wuyaqing@hqu.edu.cn (Y.W.); jianghongying@hqu.edu.cn (H.J.); merck@hqu.edu.cn (J.L.); 2Engineering Research Center of Molecular Medicine, Ministry of Education & Fujian Key Laboratory of Molecular Medicine, Xiamen 361021, China; 3Department of Food Science and Center of Excellence for the Oceans, National Taiwan Ocean University, Keelung 20224, Taiwan; judy79117@yahoo.com.tw; 4Department of Medical Research, China Medical University Hospital, China Medical University, Taichung 40402, Taiwan; 5Department of Health and Nutrition Biotechnology, Asia University, Taichung 41354, Taiwan; 6Graduate Institute of Medicine, Kaohsiung Medical University, Kaohsiung 807378, Taiwan; 7Xiamen Key Laboratory of Marine and Gene Drugs, Xiamen 361021, China

**Keywords:** *Phascolosoma esculenta*, polysaccharide, antioxidant activity, hypolipidemic effect, hepatic protective effects

## Abstract

The aims of this study were to investigate the antioxidant, hypolipidemic and hepatic protective effects of *Phascolosoma esculenta* polysaccharides (PEP). PEP was prepared from *Phascolosoma esculenta* by enzyme hydrolysis and its characterization was analyzed. The antioxidant activities of PEP were evaluated by the assays of scavenging 1,1-Diphenyl-2-picrylhydrazyl (DPPH), superoxide anion, hydroxyl radicals and chelating ferrous ion *in vitro*. It showed that PEP could scavenge radicals effectively and had favorable antioxidant activities. In the meantime, the hypolipidemic effect of PEP was investigated *in vivo* by using mice model fed with high-fat diet with or without PEP treatment. Compared with the hyperlipidemic mice without treatment, the serum levels of total cholesterol (TC) (30.1–35.7%, *p* < 0.01), triglyceride (TG) (24.5–50.8%, *p* < 0.01 or *p* < 0.05), low-density lipoprotein cholesterol (LDL-C) (49.6–56.8%, *p* < 0.01) and liver levels of TC (21.0–28.4%, *p* < 0.01), TG (23.8–37.0%, *p* < 0.01) decreased significantly, whereas serum high-density lipoprotein cholesterol (HDL-C) (47.7–59.9%, *p* < 0.01 or *p* < 0.05) increased significantly after treatment with different dosage of PEP (0.2, 0.4 and 0.8 g per kg body weight, respectively). In addition, superoxide dismutase (SOD) (10.2–22.2% and 18.8–26.9%, *p* < 0.05), glutathione peroxidase (GSH-Px) (11.9–15.4% and 26.6–30.4%, *p* < 0.05) activities in serum and liver enhanced markedly while aspartate aminotransferase (AST) (18.7–29.6% and 42.4–58.0%, *p* < 0.05), alanine transaminase (ALT) (42.7–46.0% and 31.2–42.2%, *p* < 0.05) activities, as well as the levels of malondialdehyde (MDA) (15.9–24.4% and 15.0–16.8%, *p* < 0.01 or *p* < 0.05) in serum and liver reduced markedly. Moreover, the histopathological observation of livers indicated that PEP could attenuate liver cell injury. The animal experimental results demonstrated that PEP exerted hypolipidemic and hepatoprotective roles in hyperlipidemic mice. In summary, our results above suggest that PEP might be a potential natural antioxidant and utilized as a therapeutic candidate for hyperlipidemia.

## 1. Introduction

Hyperlipidemia is one of the major endocrine diseases and has been incriminated as a contributory factor of atherosclerosis [1]. Epidemiological studies have demonstrated strong causal relations between lipid parameters level and hyperlipidemia [2,3]. The excess intake of cholesterol-rich or high-fat diet will lead to metabolic dysfunctions of lipid, resulting in hyperlipidemia. Elevated levels of total cholesterol (TC), triglyceride (TG), low-density lipoprotein cholesterol (LDL-C) and atherosclerosis index (AI) coupled with reduced serum high-density lipoprotein cholesterol (HDL-C) level are considered to be major risk factors for the development of many lipid-related diseases such as atherosclerosis or cardiovascular disease [1]. Furthermore, high levels of dietary fat could increase oxidative stress and decrease antioxidant enzyme activities. The oxidative damage could accelerate the pathogenic progress of hyperlipidemia and its complications. The conventional therapeutic modalities available for hyperlipidemia mainly include lipid lowering drugs, such as statins, fibrates and bile acid sequestrants. Although effective, long-term use of these synthetic drugs may lead to severe side effects including toxicity in muscles and liver [4]. Hence, it is of great importance to look for alternative natural hypolipidemic drugs with less or no side effect. Oxidative stress is currently suggested as a mechanism underlying hyperlipidemia. Antioxidants are increasingly recognized as potentially hypolipidemic drugs, and they play important roles in protecting against some related diseases caused by hyperlipidemia [5]. Since polysaccharide from natural materials is one of the most important compounds with various biological activities. For example, polysaccharides from marine sources are considered to possess antioxidant [6], antitumor, anti-inflammatory, anticoagulant, hypoglycemic, hypolipidemic effect and immunomodulatory ability [7,8]. Several natural marine polysaccharides display antioxidant and hypolipidemic effects, such as polysaccharides from *Enteromorpha prolifera* [9], *Ulva pertusa* [10] and sea cucumber [3,11,12], implying that polysaccharides might be developed as novel potential hypolipidemic agents.

*Phascolosoma esculenta* is a special species of Sipuncula in the Southern China. It is recognized as a kind of delicious and traditional seafood with a high nutritive value, and has long been used as a special dish with beneficial pharmacological effects for various physiological functions [13]. Literatures have indicated that *P. esculenta* exhibits a variety of biological activities, including anti-hypertensive activity [14,15], immunomodulatory activity [16], anti-fatigue [17], antioxidant [18], anti-thrombotic activity [19], is able to improve learning and memory ability [20] and stimulates the milk secretion of mice [21]. It contains many bioactive substances, such as proteins, amino acids, peptides [15,22], polysaccharides [16], fatty acids and sterols [23]. To the best of our knowledge, polysaccharides are one of the important active components of Sipuncula which reportedly have a lot of bioactivities [24], while only few have studied *P. esculenta* polysaccharide (PEP). In addition, there is scarcely any research on the effect of *P. esculenta* or PEP on hypolipidemic or hepatoprotective activities. To explore new application of *P. esculenta*, we conducted laboratory studies and our preliminary experiments showed that *P. esculenta* had potential anti-thrombotic and hypolipidemic effects, which might be closely related to the antioxidant activity. Recent researches reported that antioxidant polysaccharides of marine animal origin have been found in sea cucumber *Apostichopus japonicus* [3] and *Metriatyla scabra* [12] possessed hypolipidemic effects. These studies let us to hypothesize that PEP maybe serve as a hypolipidemic agent for hyperlipidemia. Nevertheless, its efficacy needs to be scientifically and systematically evaluated as the previous report on sea cucumber [3]. In this present study, PEP was extracted from *P. esculenta* by enzyme hydrolysis and the composition was analyzed. Subsequently, the *in vitro* antioxidant activities of PEP including 1,1-Diphenyl-2-picrylhydrazyl (DPPH) radical scavenging activity, ferrous ion chelating ability, superoxide anion scavenging capacity and hydroxyl radical scavenging effect were assessed. Finally, the effects of PEP on body weight, blood lipid, hepatic lipid and antioxidant enzyme activity were evaluated by using the high-fat diet induced hyperlipidemic model in mice, and the hypolipidemic, hepatoprotective and antioxidant activities were investigated. This study may be meaningful for improving the utilization value and application range of PEP and *P. esculenta*.

## 2. Results

### 2.1. Characterization of PEP

The water-soluble polysaccharides were obtained from *P. esculenta* by hot water extraction, composite enzyme digestion and freeze-drying [25]. It is dust-colored powder, which polysaccharide content was 31.19% and sulfate group was 0.58%. The hydrolyzed PEP was derived by 1-phenyl-3-methyl-5-pyrazolone (PMP) and then analyzed by high performance liquid chromatography (HPLC). As shown in Figure 1, HPLC indicated that PEP was composed of mannose, ribose, rhamnose, glucuronic acid, glucose, galactose, xylose, arabinose and fucose in an approximate mass ratio of 3:2:1:1.6:7.6:5.5:1.5:1:3 (Figure 1).

### 2.2. Antioxidant Activity of PEP in vitro

To confirm whether PEP has antioxidant functions, the scavenging effects of PEP on DPPH free radicals, superoxide anion radicals, hydroxyl radicals using L-ascorbic acid as positive control and the chelating ability of PEP on ferrous ion using ethylene diamine tetraacetic acid (EDTA) as position control were evaluated in vitro with biochemical tests. As shown in Figure 2, PEP showed obvious dose-dependent scavenging activities on DPPH radical, and the scavenging effect was 56.51 ± 1.89% at the concentration of 1 mg·mL^−1^ PEP (Figure 2a). Furthermore, the scavenging effects were 50% at the concentration of 1.16 mg·mL^−1^ for PEP and 0.00518 mg·mL^−1^ for L-ascorbic acid, respectively. It suggests that PEP has high performance on DPPH scavenging activity, but is not as powerful as L-ascorbic acid. Similarly, the ferrous ion chelating activity of PEP was 18.59 ± 1.21% at the concentration of 1 mg·mL^−1^ (Figure 2b), and the half-effect concentration was 6.41 mg·mL^−1^. In addition, for the superoxide radical assay and the hydroxyl radical assay, the scavenging effects were 43.30 ± 2.52% (Figure 2c) and 23.76 ± 1.18% (Figure 2d) at the concentration of 1 mg·mL^−1^, respectively, which was apparently lower than the DPPH scavenging activity. The total phenolic content was 36.63 ± 5.08 mg of gallic acid equivalent per g of PEP. For activity assay above, the activities of PEP increased obviously in a dose-dependent manner within the concentration range of 0.015625–1 mg·mL^−1^. The highest activity was DPPH scavenging radical activity, followed by superoxide anion radical activity and hydroxyl radical scavenging activity, the lowest was ferrous ion chelating activity. Moreover, there was significant difference (*p* < 0.01) between each individual activity assay at the same concentration, as those found for PEP and the position control. Overall, these data suggested that PEP had a good dose-dependent radical-scavenging activity and might be effective as an antioxidant additive, while its effectiveness varies depending on the radicals.

### 2.3. Effect of PEP on High-Fat Diet Fed Mice

#### 2.3.1. Effect of PEP on Body Weight

Hyperlipidemia is always accompanied by an increase in body weight [26]. As shown in Figure 3, the initial weight of the mice had no significant difference (*p* > 0.05) among the seven groups, indicating that the random grouping was reasonable. After 75 days of experiment, the body weights were still not significant different (*p* > 0.05) in all groups, including the normal (N) group, the negative control (NC) group, the positive control (PC) group, the PE group, the low-dose PEP (PEP-L) group, the middle-dose PEP (PEP-M) group and the high-dose PEP (PEP-H) group. These results displayed that PEP had no effect on the body weight of hyperlipidemic mice.

#### 2.3.2. Effect of PEP on Serum TC, TG, LDL-C and HDL-C Levels

The levels of lipids in serum of normal and experimental animals in each group were shown in Table 1. As shown in Table 1, after 50 days of high-fat diet feeding, the levels of serum TC, TG, LDL-C and AI in the NC group were significantly higher (*p* < 0.01) than those in the N group, whereas the serum level of HDL-C in the NC group was significantly lower (*p* < 0.01) than that in the N group, which indicated that the hyperlipidemia model in mice induced by high-fat diet was well established. Then hyperlipidemic mice were treated for 25 days, the increased serum levels of TC, TG and LDL-C were reduced significantly (*p* < 0.01 or *p* < 0.05), whereas the decreased serum level of HDL-C was elevated significantly (*p* < 0.01 or *p* < 0.05) in all treated groups than that in the NC group. Furthermore, the effect of PEP on AI was decreased significantly (*p* < 0.01) in all treated groups when compared with the NC group. In all treatment groups, there was no significant difference (*p* > 0.05) between each treatment group. Moreover, the levels in the PE group and three PEP groups were consistent with those in the PC group. However, there was no dose-dependence among three PEP groups. The results exhibited that PEP was of markedly therapeutic effect on decreasing concentration of serum TC, TG and LDL-C and increasing concentration of serum HDL-C in hyperlipidemic mice with no dose-dependence, strongly suggesting PEP has potential hypolipidemic effect against hyperlipidemia.

#### 2.3.3. Effect of PEP on Liver Index and Liver Lipids

Changes in liver index and liver lipids levels of different groups were presented in Table 2. The liver index and the liver lipids in the NC group were significantly higher (*p* < 0.01) than those in the N group, the increase rates were 22.0%, 94.0% and 87.4%, respectively. Compared with the NC group, the liver index and TC levels in liver of all treated groups were significantly reduced (*p* < 0.01 or *p* < 0.05). Similarly, the TG contents in liver of the PC, PE and PEP-M groups were significantly decreased (*p* < 0.01 or *p* < 0.05). There were no significant differences (*p* > 0.05) in the liver index and TC levels between the PE group and three PEP groups, while the TG levels in the PEP-M group were significantly lower (*p* < 0.01) than those in the PEP-L group and the PEP-H group. Compared with the NC group, the liver index, TC and TG contents in liver of the PEP-M group were decreased by 16.5%, 28.4% and 37.0%, respectively. Furthermore, there was no dose-dependence among three PEP groups and the dosage of PEP-M (0.4 g per kg body weight) was optimum. All data above demonstrated that PEP could lower liver lipids levels and protect liver against high-fat diet induced liver damage.

#### 2.3.4. Effect of PEP on Levels of Aspartate Aminotransferase (AST) and Alanine Transaminase (ALT) Activity in Serum and Liver

The effects of PEP on the AST and ALT activity in serum and liver were shown in Table 3. Apparently, levels of AST and ALT in serum and liver of the NC group were dramatically increased (*p* < 0.01) when compared with those of the N group, clearly spreading that depositing of lipid in liver resulted in damage to the liver cells. There were significant decreases in the levels of AST (*p* < 0.01) and ALT (*p* < 0.01) in serum and liver of all treatment groups without dose-dependent relationship, as the dosage of PEP-M (0.4 g per kg body weight) was optimum, when compared with the NC group. These findings clearly stated that PEP was effective to repair damaged liver cells and possessed a hepatoprotective activity.

#### 2.3.5. Effect of PEP on Antioxidant Activity of Serum and Liver

Oxidative stress plays a key role in the development and procession of hyperlipidemia. In order to explore the possible mechanism between the hypolipidemic, hepatoprotective activities and antioxidant *in vivo*, the activities of superoxide dismutase (SOD), glutathione peroxidase (GSH-Px) and the concentration of malondialdehyde (MDA) in serum and liver were investigated [26]. The results were showed in Table 4. Compared to the N group, the activities of SOD and GSH-Px were reduced significantly (*p* < 0.01 in serum and liver for SOD, *p* < 0.01 in serum and *p* < 0.05 in liver for GSH-Px), and the serum content of MDA was increased significantly (*p* < 0.01 in serum and liver) in the NC group. It indicated that continuous consumption of high-fat diet led to oxidative damage. Compared with the NC group, the PC group was found that a significantly increased in SOD activity (*p* < 0.05 in serum and *p* < 0.01 in liver) and GSH-Px activity (*p* < 0.01 in serum and *p* < 0.05 in liver) whereas a significantly decreased in MDA level (*p* < 0.01 in serum and *p* < 0.05 in liver). Likewise, the SOD and GSH-Px activities in serum and liver were significantly increased (*p* < 0.05) and the MDA level was significantly decreased (*p* < 0.05) in the PE group. The SOD activities in serum and liver were increased in PEP-L group, while were significant increased in the PEP-M and PEP-H groups (*p* < 0.05). The GSH-Px activities in serum and liver were significantly increased in the PEP-L and PEP-M groups (*p* < 0.05), while was increased in the PEP-H group. The serum MDA levels in the PEP-L group and the liver MDA levels in the PEP-M group were significantly decreased (*p* < 0.05), the serum MDA levels were significantly decreased (*p* < 0.01) in the PEP-M and PEP-H groups, while the liver MDA levels were decreased in the PEP-L and PEP-H groups. There was no significant difference (*p* > 0.05) between each treatment group for all treatment groups. Furthermore, there were no dose-dependent relationships among three PEP groups. These data (Table 4) revealed PEP could counteract the increased oxidative damage by enhancing the antioxidant enzyme activities and decreasing lipid peroxidative product levels, but no dose-dependent effect.

#### 2.3.6. Changes in Morphology and Histopathology of Liver

The morphology and histopathology were carried out to examine the effects of PEP in the livers of experimental mice, and the results were present in Figure 4.

As shown in Figure 4a, the livers of mice in the N group were dark red with sharp edges, tough textures and smooth surface, the touch was elastic, whereas the livers of mice in the NC group were cream yellow with blunt edges, friable textures and obvious swelling. Furthermore, much more yellow droplets were dispersed on the liver surface of mice in the NC group than those of either the N group or PEP-treated groups. These observations suggested that a continuous high-fat consumption might cause lipids accumulation in the liver of mice, resulting in a certain damage to the liver, even cause a certain degree of fatty liver. On the contrast, the liver morphological status of all treated groups was improved with deeper color and more smooth surface with less yellow droplets when compared with the NC group. The morphology observations were well consistent with the data of liver index, TC and TG levels in liver (Table 2). Taken all the data above, our study indicated clearly that PEP had evident inhibition on hepatic morphological changes and steatosis induced by high-fat diet. Therefore, PEP could decrease the incidences of steatosis.

As shown in Figure 4b, the changes in mice liver tissue slices of the different groups were observed by optical microscope after hematoxylin and eosin (H&E) staining. Histological examination of hepatic tissues showed that liver histology in the N group had normal hepatocyte morphology. Each hepatocyte arranged in orderly manner accompany with an abundant cytoplasm, distinct blue nuclei and well-defined cell borders (Figure 4b, N). Conversely, in the NC group, liver cells had extreme swelling and were round, and cell volume increased when compared with those in the N group. Moreover, a great number of empty vacuoles with different sizes which are generally regarded as lipid droplets by previous investigators or vesicular degeneration appeared in the cytoplasm of hepatocyte, and some nuclei disappeared. These were well in agreement with our data above liver index, TC and TG levels in liver of the NC group (Table 2), in which increase rates were 22.0%, 94.0% and 87.4%, respectively, when compared to the N group. It manifested that the accumulation of hepatic lipid droplets in the hepatic cells of the NC group.

In the contrast to those animals in the NC group, mice treated with PEP alleviated these histopathological changes markedly. Particularly, empty vacuoles (generally regarded as lipid droplets) in the liver were reduced markedly and the degeneration of the hepatocytes were obviously decreased (Figure 4b, PEP-L to PEP-H). There was a great difference in the numbers and sizes of empty vacuoles between the NC group and mice treated with PEP groups (Figure 4b, NC vs. PEP-L -PEP-H). These findings were also strongly supported by our liver lipid data above (Table 2), in which the TC and TG levels in livers of the NC group were 133% and 141% higher than the PEP-treated groups, implying that continuance in the degeneration of the hepatocytes and accumulation of hepatic lipid droplets had occurred in livers of the NC group, while for all treatment groups, histopathological changed in great deal of fat vacuoles in the liver were reduced markedly (Figure 4b, PEP-L to PEP-H) during the experimental period.

In addition, the morphology observations and histopathological analysis were consistent with data of AST and ALT activities, which were normally biochemical indicators of liver function. All data together demonstrated that PEP could attenuate the accumulation of lipid droplets in the hepatic tissue cells and prevent steatosis on high-fat diet induced hyperlipidemic mice.

## 3. Discussion

As to our knowledge, the present study was the first academic work on the hypolipidemic and hepatoprotective effects of polysaccharide from *P. esculenta* and even Sipuncula.

In this study, we have a well characterized monosaccharide of PEP (Figure 1). The monosaccharide compositions of PEP were mannose, ribose, rhamnose, glucuronic acid, glucose, galactose, xylose, arabinose and fucose, with approximate mass ratio of 3:2:1:1.6:7.6:5.5:1.5:1:3 (Figure 1). Furthermore, we discover that high-fat diet induces a marked increase in serum TC, TG, LDL-C levels and a marked decrease in serum HDL-C levels in the experimental hyperlipidemic mice (Table 1). Subsequently, successive administration of PEP produced an obvious reduction in the serum TC, TG and LDL-C levels and increase in HDL-C level by 30.1–35.7% (*p* < 0.01), 24.5–50.8% (*p* < 0.01 or *p* < 0.05), 49.6–56.8% (*p* < 0.01), 47.7–59.9% (*p* < 0.01 or *p* < 0.05), respectively (Table 1). These data clearly show that PEP attenuated the disorder of lipid metabolism, suggesting that PEP possessed an obvious hypolipidemic effect *in vivo*.

Hyperlipidemia is identified as the leading risk factor for coronary heart disease and atherosclerosis. Abnormal increases in serum TC, TG and LDL-C levels are known to develop the atherosclerosis while elevated serum HDL-C is generally regarded as protective against the development of atherosclerosis [11]. The atherosclerosis pathologic process could be slowed down or reversed by reducing serum LDL, TG and increasing serum HDL-C [3]. Moreover, the ratio of TC to HDL-C is normally used as AI, which is strong and valid indicator for monitoring and evaluating metabolic situations of cholesterol, indicating the relationship between HDL-C concentration and cholesterol-lowing activity [27]. Therefore, the lower AI is closer to the normal health state, and the effect of PEP treatment on the AI is notable. The results of this study (Table 1) display that PEP causes profound reductions in the AI of high-fat diet induced mice by 60.0% (*p* < 0.01) for PEP-L, 65.2% (*p* < 0.01) for PEP-M and 63.2% (*p* < 0.01) for PEP-H, respectively, which strongly implies that PEP might possess preventive atherosclerosis potential. Similar observation was also reported by Dong et al. [28]. However, the mechanism about the hypolipidemic effects of PEP is needed to clarify in details.

It is reported that high-fat diet as a major contributing factor in the excessive energy intake would lead to the accumulation of fat throughout many tissues, especially in liver [1]. When high-fat diet is consumed for a certain period, lipid and cholesterol are produced and stored faster in the liver cells, which cause lipid and fat metabolism disorder, as a consequence, the fatty liver of fatty degeneration or steatosis occur [1]. We therefore analyzed the effects of PEP on the development of fatty liver in this study, which was strong associated with hyperlipidemia. The evidences obtained from lab revealed that the liver index, TC and TG levels in liver were obvious decrease after treatment with PEP, which reduction rate was 16.5% (*p* < 0.01) for liver index, 28.4% (*p* < 0.01) for TC level and 37.0% (*p* < 0.01) for TG in the PEP-M group, respectively (Table 2). Since the activities of AST and ALT in serum have been used as biochemical markers for liver damage [28], lowering the liver lipid is an important approach to prevent or treatment of fatty liver [2]. Our data therefore demonstrated that a high-fat diet induced hyperlipidemia and serious liver damage as evidenced by significant elevations of AST and ALT (Table 3). Similar phenomenon has been reported on polysaccharides from *Cyclina sinensis* which induced a significant decrease of serum ALT and AST levels [29]. Furthermore, our data announced that severe injury occurred in those high-fat induced mice as indicated by massive fatty changes and ballooning degeneration (Figure 4). The greater changes in liver appearance and hepatocyte morphology appeared among the different groups. These histopathological changes (Figure 4) were observably attenuated after the treatment with PEP, implying that PEP might ameliorate the histological alteration. The continuous consumption of PEP may play a role in the development of hepatic steatosis associated with hyperlipidemia.

Accumulating literatures show that there is a close relationship between antioxidant activity and hypolipidemic activity, and have clearly presented the essential role of reactive oxygen species in the occurrence and development of hyperlipidemic disease [30]. Previous reports also supported the hypothesis that the antioxidant activity of polysaccharides played an important role in the treatment of hyperlipidemia [31]. Generally, SOD and GSH-Px are regarded as the major antioxidant enzyme that block lipid peroxidation and protect the tissue against oxidative damage, while MDA, the final product of lipid peroxidation, is considered as an important indicator of lipid peroxidation level [32]. Polysaccharides appear to be effective to improve antioxidant status, there by protect liver cells against hyperlipidemia-induced oxidative damage [33]. This epilogue is reconfirmed in our results, as shown in Table 4 and Figure 4. The hyperlipidemia mice exhibit a marked decrease in SOD, GSH-Px activities and a marked increase in MDA content in serum and liver (Table 4). After oral administration of medium dosage (0.4 g per kg body weight) PEP, there is a significantly increase in SOD, GSH-Px activities and a significantly decrease in MDA level. In addition, our antioxidant data (Figure 2) spread that PEP is of favorable antioxidant activity, which can effectively scavenge DPPH, superoxide anion, hydroxyl radicals and chelate ferrous ion in a dose-dependent manner.

When comparing the function of polysaccharides from *P. esculenta* with other reports available, the scavenging effects of PEP on DPPH and hydroxyl radicals are stronger than those reported by a recent report [34], which half-effect concentrations are 1.16 mg·mL^−1^ and 2.30 mg·mL^−1^, respectively for our study, while the half-effect concentrations were 4.4 mg·mL^−1^ and 2.6 mg·mL^−1^ [34]. The difference in data of half-effect concentrations between the two groups might be due to using different methods for processing. It is also not excluded partially due to difference in geographic distribution. For instance, we have found that geographic difference affected biochemical composition of the different populations of starfish [35,36]. When comparing the scavenging radical effect of PEP with polysaccharides from other marine species, the scavenging ability of PEP on superoxide anion (43.30%) at the concentration of 1mg·mL^−1^ is stronger than other species, for example, the ability is only 26.17% for *Sipunculus Nudus* [37]. We therefore deduce that the significant free radical scavenging activity might be the effective way of hypolipidemic effect of PEP. It is reported that there are three main mechanisms through which polysaccharides affect lipid metabolisms: (1) inhibiting of pancreatic lipase activity, (2) binding bile acids and (3) antioxidant activity [38]. In addition, it is widely recognized that the monosaccharide composition is an important factor related to the antioxidant effects of natural polysaccharides [28]. Our present work suggests that hypolipidemic activity of PEP might be at least partly-attributed to their antioxidant potential.

There were some limitations in our present study that are worth mentioning. In view of methods for various antioxidant determination, it should be very difficult to reply on a single antioxidant test model for evaluation of different antioxidant functions. There are, at least, twenty kinds of *in vitro* methods and ten kinds of *in vivo* models available so far [39]. In our present study, we just selected some of them for this study according to the advantages of each methodology and purpose of our study. For example, spectroscopic method is quite simple, while electron spin resonance should be more advanced and reliable.

In addition, it is reported that natural products could have a xenobiotic effect, and the intestinal ecosystem has been shown to be responsible for xenobiotic biotransformation [40]. Recent studies displayed that polysaccharides could reduce oxidative stress by regulating gut microbiota composition or activating gastrointestinal immune cells, and could be degraded in gastrointestinal tract, thus they could pass through the intestinal epithelial cells and enter the blood circulation [6]. As for the xenobiotic effect of PEP and its interaction with microbiota pattern should be given more attention and exploring further.

In summary, taken all results above together, PEP is of notable free radical scavenging activity both *in vitro* and *in vivo*, as well as hypolipidemic and hepatoprotective activities *in vivo*. PEP, as a food material, should be explored as a good hypolipidemic agent for hyperlipidemia therapy, and a potential strategy to attenuate the progression of atherosclerosis, liver fatty and other hyperlipidemia complications.

## 4. Materials and Methods 

### 4.1. Materials and Reagents 

The fresh *P. esculenta* was purchased from Jinjiang (Fujian, China) and identified by Professor Liu Jieqing from the Biomedical Science School of Huaqiao University. Standard monosaccharide (glucose, mannose, ribose, rhamnose, glucuronic acid, galacturonic acid, galactose, xylose, arabinose and fucose), L-ascorbic acid, EDTA, gallic acid, protease, cellulase, DPPH, 3-(2-pyridyl)-5,6-bis(4-phenyl-sulfonic acid)-1,2,4-triazine (Ferrozine), ferrous chloride, phenazin methosulfate (PMS), β-nicotinamide adenine dinucleotide (NADH), nitroblue tetrazolium (NBT), methanol, oil red O and PMP were purchased from Sigma-Aldrich Chemical Co. (Saint Louis, MO, USA). Soybean lecithin soft capsules were produced by HEALTH Co., Ltd. (Quanzhou, China). Assay kits of TC, TG, LDL-C, HDL-C, AST, ALT, MDA, GSH-Px and hydroxyl radical kit were purchased from Nanjing Jiancheng Bioengineering Institute (Nanjing, China). Total SOD activity assay kit (WST-8 method) and BCA protein assay kit were purchased from Beyotime Biotechnology Co., Ltd. (Shanghai, China). All chemicals used in this study were of analytical grade.

### 4.2. Preparation of Polysaccharides

*P. esculenta* is a high nutritional value seafood and the composition contains 85.5% moisture, 74.5% protein (dry weight), 10.8% ash (dry weight), 6.28% carbohydrate (day weight) and 2.90% fat (dry weight) [14]. The polysaccharides of *P. esculenta*, PEP were prepared according to the literature [25] with some modification. Fresh *P. esculenta* was washed to remove contaminants and dried at 28–30 °C until achieved constant weight. For extraction, dry *P. esculenta* samples were ground into fine powder (60 meshes) using dry powder grinding machine (M20 Basic, IKA Company, Staufen, Germany). Then the powdered samples (547.48 g) were weighed and extracted with boiling water at 100 ℃ for 1 h. The water extraction solutions were hydrolyzed by protease and cellulase, the enzymatic hydrolysate was centrifuged (1000 rpm, 5 min). Subsequently, the supernatant was collected, concentrated and lyophilized to get the crude polysaccharides (345.41 g), with a yield of 63.09%. The chemical composition of PEP was analyzed by the phenol-sulfuric acid method using glucose as a standard [41]. Sulfate group was determined according to the reported method [42].

The monosaccharide composition of PEP was analyzed according to the literature [43]. To analyze the monosaccharide composition, PEP was hydrolyzed with 4 M trifluoroacetic acid at 120 °C for 8 h. The hydrolyzed products of PEPs were derived with PMP according to the previous description [44]. The hydrolysate was dried by nitrogen, then was added with 1 mL of PMP methanolic solution (0.5 M) and 0.5 mL aqueous sodium hydroxide (0.3 M), and the mixture was kept for 1 h at 70 °C. After the reaction mixture was cooled to room temperature, 0.5 mL of hydrochloric acid (0.3 M) was added for neutralization, and the mixture solution was evaporated to dryness. Then the residue was added with 0.5 mL of water and 0.5 mL of chloroform and the mixture was shaken vigorously. The chloroform layer was discarded, and the aqueous layer was filtered by 0.22 μm filter membrane to be analyzed by HPLC. The monosaccharide compositions were analyzed by the Agilent 1260 HPLC with a diode array detector (DAD) at 245 nm and performed on an Agilent C18 column (250 mm × 4.6 mm,5μm). Ten microliters of the PMP derivatives were injected and eluted with KH_2_PO_4_ (0.1 M, pH 6.8) and acetonitrile in a ratio of 82%:18% (v:v) at a flow rate of 1.0 mL·min^−1^. Mannose, ribose, rhamnose, glucuronic acid, galacturonic acid, glucose, galactose, xylose, arabinose and fucose were used as reference sugars. The content assaying of monosaccharide in PEP was calculated by standard curve method.

### 4.3. Determination of in vitro Antioxidant Activities of PEP

#### 4.3.1. DPPH Free Radical Scavenging Activity

The DPPH free radical scavenging activity of PEP was measured according to the literature [45]. Samples of various concentrations (0.015625, 0.03125, 0.0625, 0.125, 0.25, 0.5, 1.0 mg·mL^−1^) and DPPH solution (0.1 mM, in methanol) were prepared and fresh on the day of each test. The 100 μL sample of various concentrations was mixed with 100 μL methanol solution of DPPH. The mixture was incubated for 30 min at room temperature in the dark, then the absorbance (A) was measured with Microplate Reader (Infinite 200, Tecan, Austria) at 517 nm against a blank. A lower absorbance of the mixture indicated a higher DPPH radical scavenging activity. Double distilled water was used as blank control, while L-ascorbic acid was used as positive control. The activity of scavenging of DPPH radical was calculated using the following equation:
(1)$$\mathrm{DPPH}\;\mathrm{radical}\;\mathrm{scavening}\;\mathrm{activity}\;\left(\%\right)=\frac{\mathrm{Acontrol}-\mathrm{Asample}}{\mathrm{Acontrol}}\times 100.\;$$
where Acontrol is the absorbance of the blank control reaction (containing all reagents except the sample) and Asample is the absorbance in the pesence of the sample.

#### 4.3.2. Ferrous Ion Chelating Activity

The ferrous ion chelating activity was measured according to the previous report [46]. The following reagents were put into a reaction tube in the following order: 150 μL of the sample (PEP of 0.015625–1.0 mg·mL^−1^), 4 μL of 2.0 mM ferrous chloride and 8 μL of 5.0 mM ferrozine. They were mixed immediately, and allowed to stand for 10 min at room temperature. Subsequently, the absorbance was monitored at 562 nm against a blank. Double distilled water was used as blank control, while EDTA was used as positive control. The ferrous ion chelating activity was calculated as follow:(2)$$\mathrm{Ferrous}\;\mathrm{ion}\;\mathrm{chelating}\;\mathrm{activity}\;\left(\%\right)=\frac{\mathrm{Acontrol}-\mathrm{Asample}}{\mathrm{Acontrol}}\times 100.\;$$
where Acontrol is the absorbance of blank control without samples and Asample is the absorbance in the presence of the sample.

#### 4.3.3. Superoxide Anion Radical Scavenging Activity

Superoxide radical scavenging activity was determined according to the method [47]. Samples with various concentrations (0.015625–1.0 mg·mL^−1^) were prepared. A 50 μL sample of different concentrations was added to a 96-well plate, then one after another to join 50μL PMS solution (120 μM), 50 μL of NADH solution (936 μM) and 50 μL of NBT solution (300 μM). After 5 min incubation in the dark at room temperature, absorbance (A) was measured at 560 nm. Double distilled water was used as blank control, while L-ascorbic acid was used as positive control. The superoxide anion radical scavenging activity was calculated using the following equation:(3)$$\mathrm{Superoxide}\;\mathrm{anion}\;\mathrm{radical}\;\mathrm{scavening}\;\mathrm{activity}\;\left(\%\right)=\frac{\mathrm{Acontrol}-\mathrm{Asample}}{\mathrm{Acontrol}}\times 100.\;$$
where Acontrol is the absorbance of blank control group (without sample) and Asample is the absorbance of sample group under 560 nm.

#### 4.3.4. Hydroxyl Radical Scavenging Activity

Hydroxyl radical scavenging activity was evaluated using the hydroxyl free radical determination kit according to the manufacture’s instruction manuals. The PEP samples were dissolved in double distilled water to obtain a serial of concentrations of 0.015625, 0.03125, 0.0625, 0.125, 0.25, 0.5, 1.0 mg·mL^−1^, and absorbance (A) was measured at 550 nm. Double distilled water was used as blank control, while L-ascorbic acid was used as positive control. The hydroxyl radical scavenging activity was calculated using the following equation:(4)$$\mathrm{Hydroxyl}\;\mathrm{radical}\;\mathrm{scavening}\;\mathrm{activity}\;\left(\%\right)=\frac{\mathrm{Acontrol}-\mathrm{Asample}}{\mathrm{Acontrol}}\times 100.\;$$
where Acontrol is the absorbance of the blank control reaction (containing all reagents except the sample) and Asample is the absorbance in the presence of the sample.

#### 4.3.5. Determination of Total Phenolic Contents

Phenolic content of PEP was estimated by the method [48]. Briefly, 1.0 mL of sample (PEP of 0.015625, 0.03125, 0.0625, 0.125, 0.25, 0.5, 1.0 mg·mL^−^^1^) was mixed with 1 mL of folin phenol reagent and incubated for 3 min; then the mixture was added with 1.0 mL 35% sodium carbonate, finally distilled water was added to make the reaction system up to 10 mL. The reaction mixture was mixed thoroughly and allowed to stand for 90 min at room temperature in the dark. Absorbance of all the sample solutions against a blank was measured at 760 nm using the spectrophotometer. Total phenolic content was expressed as milligrams of gallic acid equivalent per gram of sample (mg GAE·g^−^^1^). Calibration curve was constructed with different concentrations of gallic acid (0.02–0.12 mg·mL^−^^1^) as the standard.

### 4.4. Animals and Experimental Design

#### 4.4.1. Animals and Diets

Seventy male Kunming mice weighing approximately 20–25 g were obtained from the Animal Experimental Center of Fuzhou University (animal license No: 2015000519891, Certificate No. is SKXK. 2012-0002, Shanghai, China). The studies were conducted at the Laboratory of Pharmacology and approved by Ethics Committee of Experimental Animal Administration of Huaqiao University. All animal treatment protocols were strictly in accordance with international ethical guidelines and the National Institute of Health Guide concerning the Care and Use of Laboratory Animals. Before starting the experiments, all animals were acclimatized to the laboratory conditions for one week. They were housed in an animal room under standardized conditions (ambient temperature 25 ± 2 °C, relative humidity of 50 ± 5% and a 12/12h of light-dark cycle). They were provided with *ad libitum* food and water.

The seventy mice were randomly divided into seven parallel groups with ten mice per group, which included a normal group (N), a negative control group (NC), a positive control group (PC), a *P. esculenta* powder group (PE) and a low-dose *P. esculenta* polysaccharide group (PEP-L), a middle-dose *P. esculenta* polysaccharide group (PEP-M), a high-dose *P. esculenta* polysaccharide group (PEP-H). Mice in the N group were fed with a standard basal diet, while mice in other six groups were fed with a high-fat diet consisting 78.8% basal feed, 1% cholesterol, 10% egg yolk power, 10% lard, 0.2% bile salts (as a percentage of total kcal). After 50 days of dietary manipulation, successful model establishment was confirmed by measurement of the lipid levels. Mice in the PC group were orally administered Soybean lecithin soft capsule of 0.6 g·kg^−^^1^ body weight for 25 days. Mice in the PE group were orally administered *P. esculenta* powder of 1.35 g·kg^−^^1^ body weight for 25 days. Mice in the PEP-L group were orally administered *P. esculenta* polysaccharide of 0.2 g·kg^−^^1^ body weight for 25 days. Mice in the PEP-M group were orally administered *P. esculenta* polysaccharide of 0.4 g·kg^−^^1^ body weight for 25 days. Mice in the PEP-H group were orally administered *P. esculenta* polysaccharide of 0.8 g·kg^−^^1^ body weight for 25 days. Soybean lecithin soft capsule, *P. esculenta* powder and *P. esculenta* polysaccharides were suspended in standard saline and were intragastric administration at per kilogram body weight, respectively. Mice in the N group and the NC group were administered an equal volume of normal saline. All mice had free access to food and water. During the experiment, body weights of the mice were weighed every 5 days.

#### 4.4.2. Biochemical Analysis of Serum

At the end of the experiment (75 days), mice were weighed and blood samples were collected from the eyeballs following fasting for 12 h, and the serum was separated by centrifugation at 3500 rpm for 20 min at 4 °C. Then the serum was stored at -20 ℃ until analyzed. Levels of serum lipids including TC, TG, LDL-C, HDL-C, SOD, GSH-Px, MDA, AST and ALT were measured using commercially available kits, according to the manufacture’s instruction manuals. AI was calculated according to the following equation [28]:(5)$$\mathrm{AI}=\frac{\mathrm{TC}-\mathrm{HDL}-C}{\mathrm{HDL}-C}.\;$$

#### 4.4.3. Biochemical Analysis of Liver

Mice were sacrificed by cervical dislocation, without the use of anesthetic. The livers were excised and rinsed with standard saline, got rid of excess tissue and weighed. Parts of liver were for pathological histology and the other parts were stored at −80 °C until analyzed.

The liver morphologies were observed, and liver index was calculated according to the following equation [32]: (6)$$\mathrm{Liver}\;\mathrm{index}\;\left(\%\right)=\frac{\mathrm{Liver}\;\mathrm{weight}\;\left(g\right)}{\mathrm{Body}\;\mathrm{weight}\;\left(g\right)}\times 100.\;$$

An amount of liver tissue was mixed with phosphate buffer at the ratio 1:9 (w:v) and homogenized using a homogenizer, then centrifuged at 4 °C 3500 rpm for 10 min in a refrigerated centrifuge (H1650-W, Xiangyi Company, China). The supernatant was separated and then measured the protein content of liver tissue. The regression equation was set up by the absorbance of different concentrations standard protein. The protein concentration between 0.1–0.5 mg·L^−^^1^ and 562 nm absorbance value was a good linear relation. *Y*-axis was absorption, *X*-axis was standard protein concentration. The regression equation was y = 0.58x + 0.0769, R^2^ = 0.9952. Lipids profile including TC, TG and activity levels of SOD, GSH-Px, MDA, AST and ALT in liver tissue were determined by assay kits according to the manufacture’s instruction manuals.

#### 4.4.4. Hepatic Histology Analysis

The livers of mice were removed and a portion of livers was fixed in 10% formalin solution. Histochemical work was carried out as our previous reports [49,50]. For *in vitro* study, 3T3-L1 adipocytes stained with oil red O were subjected to observe PEP effect on changes in lipid levels before they were subjected to Microplate Reader for lipid determination. The fixed hepatic samples were dehydrated from 70% ethanol up to 100% ethanol in a graded series, then cleared in xylene and embedded with paraffin wax in molds. The embedded specimens were sectioned into 4 μm thickness using a rotary microtome and stained with hematoxylin and eosin dye, then examined under an optical microscope and photomicrographs were obtained. The areas of empty vacuoles in the liver sections were measured by a computer-image analysis system.

### 4.5. Statistical Analysis

All *in vitro* antioxidant measurements were carried out in triplicate. Data were expressed as mean values ± standard deviation. The data were assessed by SPSS software. Statistical analysis was performed by One-way analysis of variance (ANOVA) followed by Student’s *t-test* to detect inter-group differences. Statistically significant was considered at *p* < 0.05.

## 5. Conclusions

PEP, a polysaccharide isolated from *P. esculenta*, is capable of ameliorating the state of lipid metabolism in high-fat diet fed mice. Moreover, PEP possesses hepatoprotective activity against liver injury. Furthermore, PEP has antioxidant activity, as seen in the DPPH free radical scavenging assay, ferrous ion chelating activity assay, superoxide radical scavenging activity assay and hydroxyl radical scavenging activity assay. It is suggested that potent antioxidant activity of PEP may be directly or indirectly responsible for its hypolipidemic properties. Our findings for the first time revealed that PEP might be considered as an alternative functional food or pharmaceutical in the prevention and treatment of hyperlipidemia.

## Figures and Tables

**Figure 1 marinedrugs-18-00158-f001:**
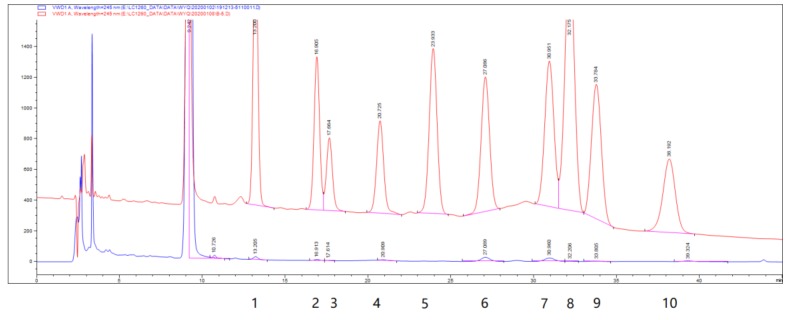
The monosaccharide composition of *P. esculenta* polysaccharide (PEP) detected by pre-column derivatization high performance liquid chromatography (HPLC). The number indicated the corresponding monosaccharide. 1, mannose; 2, ribose; 3, rhamnose; 4, glucuronic acid; 5, galacturonic acid; 6, glucose; 7, galactose; 8, xylose; 9, arabinose; 10, fucose.

**Figure 2 marinedrugs-18-00158-f002:**
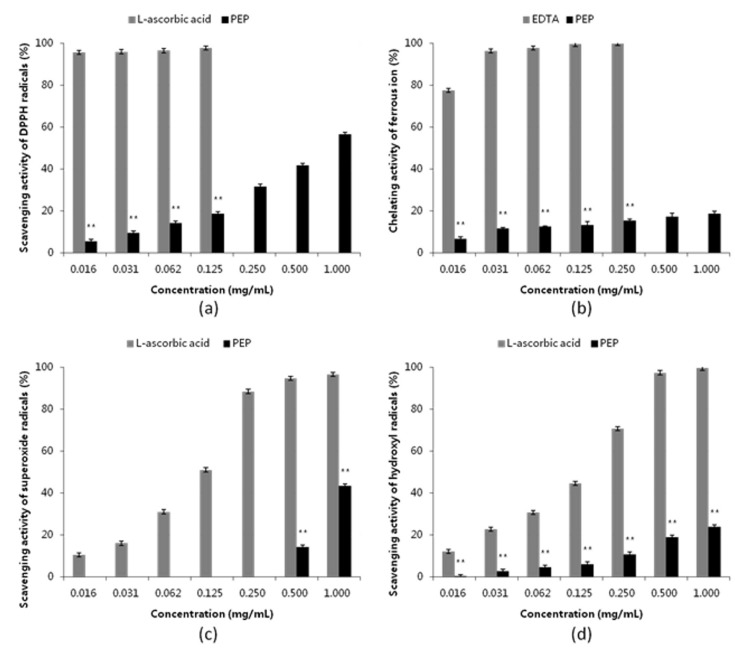
Antioxidant activities *in vitro* of PEP. (**a**) DPPH free radical scavenging activity; (**b**) Ferrous ion chelating activity; (**c**) Superoxide anion radical scavenging activity; (**d**) Hydroxyl radical scavenging activity. L-ascorbic acid and EDTA used as positive control. Data are expressed as a mean ± standard deviation (*n* = 3). ** *p* < 0.01: compared with the positive control at the same concentration.

**Figure 3 marinedrugs-18-00158-f003:**
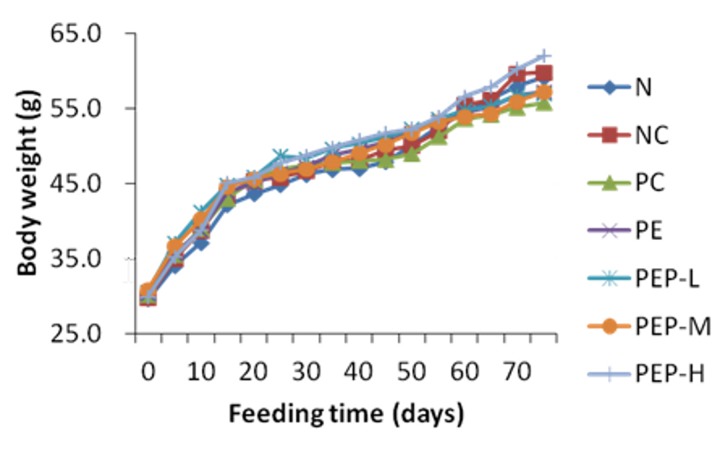
Changes in body weight of the mice after fed with different foods for 75 days. The normal (N) group was fed the basic diet; the negative control (NC) group was fed with a high-fat diet with 78.8% basal feed, 1% cholesterol, 10% egg yolk power, 10% lard, 0.2% bile salts; the positive control (PC) group was fed with a high-fat diet feed plus Soybean lecithin soft capsule 0.6 g·kg^−1^ body weight; the PE group was fed with a high-fat diet feed plus PE powder 1.35 g·kg^−1^ body weight; the low-dose PEP (PEP-L) group was fed a high-fat diet feed plus PEP 0.2 g·kg^−1^ body weight; the middle-dose PEP (PEP-M) group was fed with a high-fat diet feed plus PEP 0.4 g·kg^−1^ body weight; the high-dose PEP (PEP-H) group was fed with a high-fat diet feed plus PEP 0.8 g·kg^−1^ body weight.

**Figure 4 marinedrugs-18-00158-f004:**
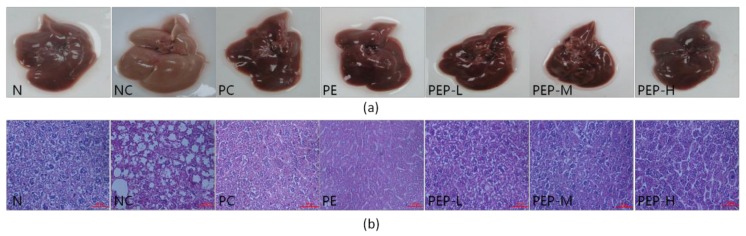
Changes in morphology and histopathology of liver. (**a**) Effect of PEP on liver morphologies in different groups, (**b**) Effect of PEP on histological structure of livers in different groups. Haematoxylin and eosin staining, Original mignification 400×.

**Table 1 marinedrugs-18-00158-t001:** Effects of PEP on serum lipid levels (*n* = 10, mean ± standard deviation).

Group	TG (mmol·L^−^^1^)	TC (mmol·L^−^^1^)	LDL-C (mmol·L^−^^1^)	HDL-C (mmol·L^−^^1^)	AI
N	1.19 ± 0.27	4.02 ± 0.66	0.27 ± 0.11	2.15 ± 0.68	1.09 ± 0.91
NC	2.52 ± 0.66 ^##^	7.79 ± 1.70 ^##^	1.08 ± 0.45 ^##^	0.89 ± 0.23 ^##^	8.40 ± 3.15 ^##^
PC	1.56 ± 0.57 **	5.35 ± 0.99 **	0.47 ± 0.24 **	1.47 ± 0.38 **	2.86 ± 1.21 **
PE	1.53 ± 0.32 **	5.40 ± 0.74 **	0.45 ± 0.35 **	1.38 ± 0.54 *	3.25 ± 1.14 **
PEP-L	1.90 ± 0.55 *	5.45 ± 1.41 **	0.47 ± 0.41 **	1.31 ± 0.39 *	3.36 ± 1.27 **
PEP-M	1.25 ± 0.33 **	5.00 ± 1.02**	0.55 ± 0.15 **	1.38 ± 0.45 **	2.92 ± 1.35 **
PEP-H	1.24 ± 0.49 **	5.35 ± 1.07 **	0.53 ± 0.33 **	1.42 ± 0.40 **	3.09 ± 1.47 **

N: the normal group; NC: the negative control group; PC: the positive control group (Soybean, 0.6 g·kg^−1^ body weight); PE: the PE powder group (1.35 g·kg^−1^ body weight); PEP-L: the PEP low-dosage group (0.2 g·kg^−1^ body weight); the PEP-M: PEP middle-dosage group (0.4 g·kg^−1^ body weight); PEP-H: the PEP high-dosage group (0.8 g·kg^−1^ body weight). ^##^
*p* < 0.01: compared with the normal group (N); * *p* < 0.05 and ** *p* < 0.01: compared with the negative control group (NC)

**Table 2 marinedrugs-18-00158-t002:** Effect of PEP on liver index and lipid profiles (*n* = 10, mean ± standard deviation).

Groups	Liver index (%)	TC (mmol·gprot^−^^1^)	TG (mmol·gprot^−^^1^)
N	3.63 ± 0.38	0.078 ± 0.004	0.051 ± 0.006
NC	4.43 ± 0.20 ^##^	0.152 ± 0.008 ^##^	0.096 ± 0.007 ^##^
PC	3.66 ± 0.26 **	0.101 ± 0.02 **	0.060 ± 0.009 **
PE	3.75 ± 0.35 **	0.119 ± 0.014 **	0.066 ± 0.006 *
PEP-L	4.07 ± 0.35 *	0.109 ± 0.012 **	0.070 ± 0.005
PEP-M	3.70 ± 0.43 **	0.108 ± 0.013 **	0.061 ± 0.005 **
PEP-H	3.65 ± 0.50 **	0.120 ± 0.015 **	0.074 ± 0.009

^##^*p* < 0.01: compared with the normal group (N); * *p* < 0.05 and ** *p* < 0.01: compared with the negative control group (NC).

**Table 3 marinedrugs-18-00158-t003:** Effect of PEP on Aspartate Aminotransferase (AST) and Alanine Transaminase (ALT) in the serum and liver (*n* = 10, mean ± standard deviation).

Group	Serum AST (U·L^−^^1^)	Serum ALT (U·L^−^^1^)	Liver AST (U·gprot^−^^1^)	Liver ALT (U·gprot^−^^1^)
N	13.98 ± 0.80	4.27 ± 0.73	3.16 ± 1.37	5.08 ± 0.39
NC	28.68 ± 1.99 ^##^	14.55 ± 1.51 ^##^	12.15 ± 1.46 ^##^	24.50 ± 1.35 ^##^
PC	17.08 ± 1.96 **	9.71 ± 0.98 **	5.36 ± 0.91 **	8.92 ± 0.92 **
PE	20.67 ± 2.03 **	8.76 ± 0.94 **	6.84 ± 0.85 **	15.93 ± 1.23 **
PEP-L	23.32 ± 2.00 **	8.34 ± 1.02 **	6.25 ± 0.94 **	16.84 ± 1.01 **
PEP-M	22.66 ± 2.66 **	8.08 ± 0.76 **	5.10 ± 0.88 **	14.15 ± 1.46 **
PEP-H	20.19 ± 2.10 **	7.86 ± 0.92 **	7.00 ± 0.97 **	15.62 ± 1.07 **

^##^*p* < 0.01: compared with the normal group (N); ** *p* < 0.01: compared with the negative control group (NC).

**Table 4 marinedrugs-18-00158-t004:** Effect of PEP on superoxide dismutase (SOD), glutathione peroxidase (GSH-Px) activities and malondialdehyde (MDA) levels (*n* = 10, mean ± standard deviation).

Group	Serum SOD (U·mL^−^^1^)	Serum GSH-Px (U·mL^−^^1^)	Serum MDA (nmol·mL^−^^1^)	Liver SOD (U·gprot^−^^1^)	Liver GSH-Px (U·mgprot^−^^1^)	Liver MDA (nmol·mgprot^−^^1^)
N	51.95 ± 8.14	2798.36 ± 186.85	10.37 ± 0.94	28.35 ± 4.91	371.92 ± 104.69	2.65 ± 0.35
NC	41.77 ± 7.20 ^##^	2317.42 ± 330.65 ^##^	13.72 ± 1.87^##^	19.31 ± 3.78 ^##^	268.11 ± 35.81 ^#^	3.47 ± 0.58 ^##^
PC	50.14 ± 6.28 *	2691.38 ± 119.74 **	10.96 ± 0.75**	24.57 ± 4.05 **	347.23 ± 84.93 *	2.90 ± 0.61 *
PE	49.55 ± 5.54 *	2607.26 ± 209.84 *	11.29 ± 1.95 *	24.20 ± 4.15 *	352.80 ± 99.33 *	2.91 ± 0.51 *
PEP-L	46.03 ± 5.99	2673.23 ± 348.06 *	11.53 ± 1.49 *	22.93 ± 4.49	345.59 ± 92.85 *	2.92 ± 0.64
PEP-M	51.02 ± 8.09 *	2655.42 ± 381.07 *	10.37 ± 0.97 **	24.50 ± 4.39 *	349.73 ± 111.13 *	2.89 ± 0.61 *
PEP-H	50.56 ± 6.50 *	2594.17 ± 348.20	10.57 ± 0.92 **	23.97 ± 3.49 *	339.52 ± 129.96	2.95 ± 0.73

^##^*p* < 0.01: compared with the normal group (N); * *p* < 0.05 and ** *p* < 0.01: compared with the negative control group (NC)

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
