# Peer review of "Antioxidant, Hypolipidemic and Hepatic Protective Activities of Polysaccharides from *Phascolosoma esculenta"

_marinedrugs, 2020, doi:10.3390/md18030158_

Round 1
Reviewer 1 Report
Comments to the Authors of manuscript number: marinedrugs-736247 entitled Antioxidant, Hypolipidemic and Hepatic Protective Activities of Polysaccharides from Phascolosoma esculenta.
In the present paper I have reviewed the antioxidant, hypolipidemic and hepatic protective effects of Phascolosoma esculenta polysaccharides (PEP) was investigated. Phascolosoma esculenta is economic important species in the fishery and aquaculture in southeast China.
The manuscript includes a lot of understatement. It needs major revision.
- L 79 – using for the first time a abbreviation it should be explained. Abstract is not considered in this point of view.
- L 91 – reference should be given
- L 45 – not coma but the word of “or”
- L 109 –why by capital letter?
- L 83 – reference should be given
- L 173 – English is poor
- L 209 - these are speculations. Proper analysis is needed.
- L 211- made?
- L215 – what staining?
10.L 220-225- was made the staining for fat ? where is a proof?
- for liver analysis, the histomorphometry is needed. Not only description.
The histopathological description is poor without proper parameters.
12.L 338- the reference should be related to Phascolosoma esculenta not to Momordica Charantina.
- L 340- what is “constant weight”?
- L 343 – English. “were hydrolysis”
This description is difficult to understood by poor English.
- L 347 – this modification should be described!
- L 348-349 It should be described clearly.
- Section from L 337 – the composition should be given. Not only ingredients which were analysed.
- L 372, L377 – “The Ferrous ion chelating” or “The ferrous ion-chelating”
- L 381 – this modification should be given.
- L 459 – this part should be described clearly. In the manuscript is a lack of the reference of number of 48.
The last reference “Shi, J.; Zheng, D.X.; Liu, Y.X.; Sham, M.H.; Tam, P.; Farzaneh, F.; Xu, R.A. Overexpression of soluble TRAIL induces apoptosis in human lung adenocarcinoma and inhibits growth of tumor xenografts in nude mice. Cancer Res. 2005, 65(5), 1687-1692.” relates to human lung!
The methods of staining and analysis should be described. Parameters which were analysed should be given.
Anatomical part of liver, from which a fragment for further analysis was given should be described.
Fibrosis should be checked.
Reviewer 2 Report
Dear authors,
After the review process, I have several comments: you should insert numerical data in the abstract; you should include a standard for each activity determined in figures and more details in Materials and Methods sections; you should include statistical relevance in figures; you should present a correlation between the methods and comment the possible limitation of the study; you should insert a significance of the results because natural products could have a xenobiotic effect?, please comment this possibility; you should insert data related to this subject published in mdpi journals, for example: https://doi.org/10.3390/biomedicines8020039, where interactions with microbiota pattern could mediate the clinical significance.
Best regards!
Reviewer 3 Report
The manuscript is interesting enough to be acceptable for publication in Marine Drugs.
However, the authors should be very careful to the following points.
1, The authors evaluated the DPPH scavenging activity based on the spectroscopic analysis. The quenching activity of the compounds was calculated based on the decrease ratio of the absorbance. If 100 % quenching takes place, the base line (absorbance) becomes 0 ? It is not clear.
It is highly desirable to use the ESR machine because it is obvious to show the radical quenching.
2, It is also the case of superoxide scavenging activity using NBT.
3, The authors carried out many experiments, however the statistical analysis lacks accuracy.
The authors always showed statistical difference between group PE, PEP-L, PEP-M, PEP-H versus N or NC.
It is also quite important to show the dose dependency of PEP by showing the statistical difference.
Based on these facts, the reviewer does not recommend this manuscript acceptable for publication in the present form.
Round 2
Reviewer 1 Report
this manuscript in this form can be accepted
Reviewer 2 Report
Dear Authors,
I do not have any other comments.
Best regards!
Reviewer 3 Report
In the revised version of the manuscript, the authors made significant revisions, that might strengthen the manuscript.
Although the manuscript has still some drawbacks, however, it will be clarified in the future version as the authors wrote.